# Studies on Lotus Genomics and the Contribution to Its Breeding

**DOI:** 10.3390/ijms23137270

**Published:** 2022-06-30

**Authors:** Huanhuan Qi, Feng Yu, Jiao Deng, Pingfang Yang

**Affiliations:** State Key Laboratory of Biocatalysis and Enzyme Engineering, School of Life Science, Hubei University, Wuhan 430062, China; qihuanhuan0911@163.com (H.Q.); yufeng@hubu.edu.cn (F.Y.); ddj613@163.com (J.D.)

**Keywords:** lotus, genome, variant, germplasm, breeding, omics

## Abstract

Lotus (*Nelumbo nucifera*), under the Nelumbonaceae family, is one of the relict plants possessing important scientific research and economic values. Because of this, much attention has been paid to this species on both its biology and breeding among the scientific community. In the last decade, the genome of lotus has been sequenced, and several high-quality genome assemblies are available, which have significantly facilitated functional genomics studies in lotus. Meanwhile, re-sequencing of the natural and genetic populations along with different levels of omics studies have not only helped to classify the germplasm resources but also to identify the domestication of selected regions and genes controlling different horticultural traits. This review summarizes the latest progress of all these studies on lotus and discusses their potential application in lotus breeding.

## 1. Introduction

Food shortage has become a rising challenge with the increase of the world’s population and decrease of natural resources. It is incredibly significant to breed crops with high yields, of good quality, and high-stress resistance to ascertain food security since crops provide a staple food supply for the world. To achieve this, it is necessary to obtain a deeper understanding of the crops’ genetic background, especially their genome information. Since the first flower plant, *Arabidopsis thaliana* genome was sequenced in 2000 [1], more and more plant whole genomes have been sequenced and deposited in databases, which are available to the public [2], including *Nelumbo* genome database (http://nelumbo.biocloud.net/nelumbo/home) [3]. Third-generation sequencing, which can produce long sequence read, has shown its advantages over next-generation sequencing (NGS) in generating high continuity reference genome assemblies [4]. 

Genomics is the cornerstone of breeding, and studies based on whole-genome sequencing and genome-wide association study have greatly driven forward genomics-assisted breeding in many crops [5,6]. Cloning and functional analysis of genes associated with important agronomic traits in rice (*Oryza Sativa*), soybean (*Glycine max*), and tomato (*Solanum lycopersicum*) have also demonstrated that high-quality genomes are prerequisite to clarify variations in each species [7,8,9,10,11]. However, population genetic analysis relied on a single reference genome that lost variant information, especially in the highly polymorphic region. Pan-genome contains the totality of genome sequence information of the target species and covers more comprehensive variant information. Pan-genomes have been constructed in many plants, such as rice, maize, brassica, and soybean, and applied to identify causal genes [12,13,14,15]. Pan-genome or graph pan-genome is obtaining new references along with the upgrading of sequencing. The information on genome maps, domestication, improvement-related genes, and regulation pathway promotes the understanding of plant evolution and accelerate breeding [15]. 

Lotus is one of the relict plants retaining the original morphology of its ancestors, as well as *Ginkgo biloba*, *Liriodendron*, and *Metasequoia glyptostroboides*. It belongs to the *Nelumbo* genus of the Nelumbonaceae family, which includes two species, namely Asian lotus (*Nelumbo nucifera* Gaertn.) and American lotus (*Nelumbo lutea* Pear.). The two species are named for their different geographical distributions. Asian lotus is mainly distributed in Asia and the north of Oceania, while American lotus is distributed in North America and South America. The plant morphology differs between them. Asian lotus is a tall plant, with oval leaves and seeds, and red or white flower colors, whereas American lotus is a short plant, nearly round and with dark green leaves, spherical seeds, and yellow flowers [16]. There is no strict reproductive isolation between them, and the life cycles are similar at about five months. Asian lotus is commonly called lotus and has more than 3000 years of cultivation history as a horticultural crop [17]. Lotus seeds and rhizomes have rich nutritional value and unique health-care function. Lotus seeds contain starch, proteins, amino acids, polysaccharides, polyphenols, alkaloids, and mineral elements. Lotus rhizome has a high vitamin C content. During the long period of domestication and artificial selection, about 4500 lotus cultivars have been obtained up till the present [18]. These cultivars have been planted to produce edible vegetables, snacks, beverages, restorative materials, and ornamental flowers, which impact human life and economic development. The lotus industry is also important for rural revitalization in the Yangtze River, Pearl River, and Huang Huai river basins. The cultivated lotus is generally divided into rhizome lotus, seed lotus, and flower lotus based on their different usage. The notable feature of the rhizome lotus is the enlarged rhizome but with few flowers. It can be divided into power and crisp type according to the taste of the rhizome. Different varieties were bred to meet the taste of the different regions of people or for further usage. The main breeding goal of rhizome lotus is to improve the yield and quality of the rhizome. Seed lotus is mainly for lotus seed production, with high yield, good quality, and disease resistance being the breeding goals. Flower lotus is preferred for ornamental use, and it has distinct flower colors and shapes. During long cultivation, ornamental lotus with different flower morphologies were obtained, including few-petaled, double-petaled, petaloidy, and thousand-petalled flowers. Red, pink, yellow, and white are the main flower colors. Currently, the breeding objective is mainly aimed at flower shape and color, yield or quality of lotus seed and rhizome, and wide adaptability. 

As a basal eudicot species, lotus plays an essential role in studying plant evolution and phylogeny. It is adapted to the aquatic environment, while its relatives are shrubs or trees living on land. Water lily lies at the phylogenetic position of the base angiosperm and has similar living conditions and flowers. However, its genomes are vastly different [19]. Lotus has unique features such as water-repellent self-cleaning function, multi-seed production, and flower thermogenesis, which may relate to flower protogyny or provide a warm environment for pollination [16]. Because of its importance in plant phylogeny and wide application, lotus has gained increasing attention from the scientific community. Since the release of the first version of two lotus reference genomes [20,21], genome-based investigations have been conducted continuously. Subsequently, the high-resolution genetic map and BioNano optical map were applied to improve the accuracy and assembly of the lotus genome [22]. A hybrid assembly was completed using PacBio sequencing data and previously published short reads [23]. High-quality genome assembly of “Taikonglian NO. 3” and American lotus genome were also recently generated [24,25]. High-throughput re-sequencing of different lotus cultivars has been utilized to identify numerous molecular markers, promoting marker-assisted selection. Moreover, “omics” approaches such as transcriptomics, proteomics, and metabolomics were applied in elucidating molecular regulatory networks of yield, quality, and response to stress in lotus. Here, we briefly review the latest progress of studies on the lotus genome, and how genome information could be used in lotus breeding. Meanwhile, the existing challenge and potential prospects are also discussed. 

## 2. Sequencing, Assembly and Annotation of Lotus Genome

Lotus occupies a crucial phylogenetic position in flowering plants. The high-quality reference genome of lotus plays a vital role in studying the origin of eudicot and lotus molecular breeding. In the last decade, some lotus varieties were sequenced by different platforms, which resulted in a different version of the genome assembly and annotation (Table 1). Based on NGS, a wild lotus, “China antique (CA)”, was successfully sequenced and assembled [20]. The total sequenced genome length of “CA” is 804 Mb, of which 543.4 Mb (67.6%) were anchored to nine megascaffolds. The contig N50 was 38.8 Kb and the scaffold N50 was 3.4 Mb. The heterozygosity of “CA” genome is 0.03%, and the repetitive sequence is about 57%. A total of 26,685 protein-coding genes were predicted, with the average length of a gene being 6561 bp. Simultaneously, another wild strain of lotus, “Chinese Taizi” was assembled through NGS technology. The final assembled genome size is 792 Mb with the contig N50 39.3 Kb and scaffold N50 986.5 Kb [21]. The length of transposable elements is 392 Mb (49.48%), and 36,385 protein-coding genes were annotated. One WGD event -λ in lotus instead of the paleo-hexaploid arrangement (γ WGD) event that occurred in core eudicots was predicted [20,21]. These two genomes were further anchored to eight pseudo-chromosomes by constructing a higher resolution genetic map and physical maps [22].

With the advent of a new sequencing platform, the genome of “CA” was re-assembled using 11.9 Gb long-read data from PacBio Sequel, and 94.2 Gb previously sequenced short-read data [23]. The new assembly of “CA” is 807.6 Mb with the contig N50 being 484.3 Kb, which has significantly increased the quality of the genome. The ratio of repetitive sequence (58.5%) was similar to the first version. Moreover, a cultivated lotus, “Taikonglian NO. 3”, was also assembled using the Oxford Nanopore sequencing platform (57.9 Gb raw data) with the contig N50 being 5.1 Mb, and eight chromosomes were anchored based on high-throughput chromatin conformation capture (Hi-C) data [24]. Another lotus species, American lotus, was recently assembled using PacBio RSII (74.6 Gb raw data) and Hi-C (50.32 Gb raw data), and the total length is 843 Mb while contig N50 is 1.34 Mb [25]. These data demonstrate that long-read sequencing technology has greatly improved the quality of the lotus genome. The successful assembly of the genome in Asian lotus, including wild and cultivar varieties, and American lotus will assist the investigation of functional genomics as well as molecular breeding in lotus. 

## 3. Study on the Potential Adaptive Evolution and Domestication of Lotus

The availability of lotus reference genome information has facilitated the resequencing of different lotus germplasms. Several studies were conducted on how the lotus genome was subjected to adaptive evolution and artificial selection. Although it is known that there are only two species of lotus, namely Asian lotus and American lotus, except for the difference in flower color, their plant architecture and morphology are very similar. Based on molecular phylogeny analysis, significant genetic differentiation between American and Asian lotus was verified [25,26,27,28]. De-novo deep sequencing of the American lotus showed that its genome size is 843 Mb, and an approximate 81% repeat sequence was identified (Table 1), which is larger than the genome of Asian lotus. It is interesting to investigate the dramatic difference in repeat sequence between them because most protein-coding genes show a one-to-one synteny pattern. A total of 29,533 structure variations (SVs) were detected between two lotus species, with the SV-associated genes overexpressed in ‘regulation of mitotic cell cycle’, and ‘protein transporter activity’ [25]. Meanwhile, this study also showed that the selection on an *MYB* gene might contribute to the color difference between Asian and American lotus [25]. It is still an open question about when the two species diverged during the evolution and how they could keep high similarity in the independent geographical evolution. The wild lotus is distributed widely worldwide and maintains higher genomic diversity than cultivated lotus. Tropical and temperate lotus are the two ecotypes of Asian lotus. The comparison of the genome of these two ecotypes showed that a total of 453 genes were subjected to selection, including *cyp714a* genes that may relate to rhizome morphogenesis and a 10-Mb region in chromosome 1 that might play key roles in environmental adaption; including a homolog gene of *at5g2394* in *Arabidopsis* encoding an acyltransferase protein [24]. By comparing their expressional patterns, the genes encoding granule-bound starch synthases, storage organ development, *COSTAN-like* gene family, vernalization, as well as cold response genes may relate to ecotypic differentiation [26].

It is very important to know the genetic backgrounds of parental lines in breeding. The origin, classification, and evolution of cultivated lotus were investigated through population re-sequence analysis. A total of 18 lotus accessions, including categories of American, seed, rhizome, flower, wild, and Thai lotus, were re-sequenced, based on which phylogenetic tree was constructed. The results indicated that the rhizome lotus had a closer relationship with wild lotus. In contrast, seed and flower lotus were admixed [26], which could be supported by re-sequencing of an enlarged population containing 296 accessions of different germplasm (58 wild, 163 rhizome, 39 flower, 32 seed lotus varieties) [28]. Further re-sequencing of 69 lotus accessions showed that flower lotus might mix with rhizome or seed lotus [27]. All studies showed a low genetic variation in rhizome lotus, while higher genetic diversity in seed lotus. The origin of different subgroups is controversial, which is possible because the same accession of lotus has other names which were then divided into different subgroups by various people. Based on this genomic diversity, the potential domestication signals of cultivar subgroups could be speculated because the selected genomic regions had lower nucleotide diversity. When subgroups of seed, rhizome, and flower lotus were compared with wild lotus subgroup, a total of 1214, 95, and 37 artificially selected regions containing 2176, 77, and 24 genes were identified in seed, rhizome, and flower lotus, respectively [27]. Several of these selected genes were involved in key developmental processes associated with different organs. For example, a *SUPER-MAN like* gene affecting seed weight and size and a *legumin A-like* gene involved in storage protein synthesis were identified in the subgroup of seed lotus, while an *expansin-A 13-like* gene was identified in the subgroup of rhizome lotus [27]. These specifically selected genes controlling agronomic traits in different subgroups are also possible targets for lotus breeding. Meanwhile, different types of molecular markers have been developed, which may further facilitate the clarification of the relationship between different subgroups and maker-assisted breeding of new lotus varieties [29,30,31,32,33,34,35,36,37,38].

## 4. Identification of Genes with Potential Application in Lotus Breeding

As the largest aquatic vegetable in China, lotus is mainly bred through traditional cross-breeding and physical and chemical mutation as supplementation, based on which thousands of varieties have been obtained [39]. However, the selection of high-quality varieties was mainly based on the breeders’ experience, because the mechanisms underlying each economic trait remained unclear. With the development of genomics and molecular genetics of the lotus, genome-based breeding is gradually becoming an effective method for lotus. Causal genes regulating essential traits, such as flower color and shape, rhizome yield, and seed quality, have been widely studied.

Flower color, shape, and flowering time are important traits that determine the ornamental value of lotus. There are three different colors in lotus, red and white in Asian lotus and yellow in American lotus. The red color in Asian lotus is determined by the contents of anthocyanin [40,41], which is controlled by key enzyme encoding genes, and their regulating transcription factors (TFs) such as *MYB*, *basic-Helix-Loop-Helix* (*bHLH*), *WD40* in its biosynthetic pathway. Among all the enzyme encoding genes in this pathway, *NnANS* and *NnUFGT* seem to be the decisive two genes [42,43]. Several TFs including 5 *MYB*, 2 *bHLH*, and one *WD-repeat* genes, may be involved in the regulation of anthocyanin biosynthesis in lotus based on a transcriptome analysis [43]. Among them, a *bHLH* gene *NnTT8* was verified to regulate anthocyanin biosynthesis [44], whereas the yellow color of American lotus is determined by carotenoid, and no anthocyanin was detected [25,45]. Further analysis indicated that the difference in the coding region between *NnMYB5* (Genbank accession, KU198697) and *NlMYB5* (Genbank accession, KU198698) is the main reason for the different colors in the two species. Flower morphology is another factor that determines the ornamental value of lotus. Flower development is controlled by intricate gene-regulatory networks, and many vital genes that control flowering time have been identified in flowering plant species. However, the molecular regulation mechanism has not been well characterized in lotus. Comparative transcriptomic analysis of different bud development stages in temperate and tropical lotus identified 147 lotus flowering-time associated genes that participate in photoperiod, gibberellic acid and vernalization pathways [46]. The *MADS-box* TFs are widely involved in plant growth and development. A total of 44 *MADS-box* genes were identified in lotus, and based on the selected candidates, *NnMADS14* (*SEPALLATA3* (*SEP3*) homolog gene) was identified to be related to floral organogenesis in lotus [47]. Lotus possesses distinct types of flower morphology, and the floral organ petaloid phenomenon is universal. Comparative transcriptomic analysis identified many hormonal signal transduction pathway genes and *MADS-box* genes; *AGAMOUS*(*AG*) was predicted as the candidate which was gene related to carpel petaloidy [48,49]. Genome-wide DNA methylation analysis showed that different flower organs exhibited different methylation levels, while *plant U-box* (*PUB33*) homolog gene might play crucial roles in the stamen petaloid [50]. Furthermore, *NnFTIP1* was proven to interact with *NnFT1* and regulate the flowering time in lotus [51]. 

The rhizome is the main edible part of lotus. It is important to explore the mechanisms underlying rhizome formation and expansion in rhizome lotus breeding. Comparative transcriptomic and proteomic analyses focusing on rhizome development have been conducted to dig out the key genes and pathways critical for the crucial physiological process [52,53,54]. Furthermore, re-sequencing of the natural and genetic F_2_ populations has also identified several genetic regions and candidate genes that might be involved in lotus rhizome enlargement [55]. A systematic analysis was conducted on one candidate gene *CONSTANS-LIKE 5* (*COL5*). Functional analysis in the potato system indicated that *NnCOL5* might be positively associated with rhizome enlargement by regulating the expression of *CO-FT* genes and the GA signaling pathway [56]. In addition, one SNP was identified in another candidate gene *NnADAP* of *AP2* subfamily, which is closely associated with rhizome enlargement phenotype and the soluble sugar content [57]. There is a big difference between temperate and tropical lotus, especially the rhizome’s morphology. Many genes were highly differentiated between them, such as *APL* homologs and granule-bound starch synthases genes [26]. Temperate lotus is distributed at 20° north latitude and shows a significant annual growth cycle, whereas tropical lotus is distributed south of 17° north latitude and exhibits perennial growth. Asian wild lotus can be further divided into temperate, subtropical, and tropical types and is distributed in northeast China, the Yangtze River and Pearl River Basin, Thailand and India [27]. Different lotus groups are subject to different selection pressure, such as light, temperature, UV, and soil types. The genes underlying selection were discussed by integrating population genetics and omics data. Several genes related to photosynthesis and DNA repair were selected, such as NAD + ADP ribosyltranferase, 8-oxoguanine-DNA glycosylase 1 and DNA polymerase epsilon subunit B2. The *vacuolar iron transporter* (*VIT*) family gene, *nodulin-like 21* gene encoding vacuolar iron transporter, may be related to metal ion metabolism [24]. The homolog gene of *Arabidopsis VIN3* in lotus was predicted to be related to flowering time and dormancy, with higher expression in temperate lotus than in tropical lotus [26]. 

Lotus seeds are rich in nutrients and functional compounds such as alkaloids, flavonoids, and polyphenols [58,59]. They are consumed “as both food and medicine” [60]. It is essential to increase the yield and nutrition of lotus seed. The main factors determining lotus seed yield are the seed size and the number of lotus seeds per seedpod. Transcriptome analysis on the cotyledon of “CA” and “Jianxuan-17 (JX-17)” seeds at different developmental stages identified 8437 differentially expressed genes (DEGs). Many DEGs are involved in the brassinosteroid biosynthesis pathway, and further analysis predicted two *AGPase* genes as candidate genes affecting lotus seed yield [61]. It seems that phytohormones are involved in lotus seed development. A combination of metabolomic and proteomic methods revealed that 15 DAP (Day After Pollination) was a switch time point from the physiological active to the nutrition accumulation stage [62]. Starch is the primary nutritional component in mature lotus seed [63]. Its contents and the proportion of amylose and amylopectin could largely determine the nutritional value and taste of lotus cotyledon, respectively. ADP-glucose pyrophosphorylase (AGPase) plays an important role in regulating starch biosynthesis. The evolution of *AGPase* genes experienced a purification selection, and *NnAGPL2a* and *NnAGPS1a* were the candidate genes related to starch content [64]. Starch branching enzyme (SBE) genes are key regulatory genes during starch synthesis, and *NnSBEI* and *NnSBEIII* were identified as related to the chain length of amylopectin in lotus [65]. In addition, comparative metabolomics between wild germplasm “CA” and domesticated cultivar “JX-17” indicated that the seed yield and the content of metabolites showed trade-offs [66]. For nutritional and medicinal values of lotus seed, the metabolomics-assisted strategy might be applied in lotus breeding in the future [67]. Seed dormancy is one of the domestication traits. The classical stay-green *G* gene controlling seed dormancy was cloned in domestication and as improvement genes in soybean, rice, and tomato. G gene interacts with *NCEDS* and *SPY* and in turn, regulates abscisic acid (ABA) synthesis [68]. *NnDREB1* and *NnPER1* were identified from lotus and may be involved in the ABA signal transduction pathway and then modulate longevity and dormancy [69]. 

Except for the above breeding objectives, there are other diversified breeding objectives, such as resistance to submerging and high antioxidant content. Lotus has evolved novel features to adapt to aquatic lifestyle. Many putative copper-dependent proteins, especially *COG2132* gene family, expand in lotus and form a separate phylogenetic clade having functions distinct from *Arabidopsis* [20]. Research has shown that although lotus grows in water, it is actually “afraid” of water. A time-course submergence experiment and RNA-seq analysis showed lotus has a low tolerance to complete submergence stress, and took two major strategies to cope with submergence stress in different stages. In the early stage (3~6 h) it initiates a low oxygen “escape” strategy (LOES), with the rapid accumulation of ethylene, rapid elongation of petioles, and significantly increases the density of aerenchyma and *ERF-VII* genes while lotus innate immunity genes become elevated; In the later stage (24~120 h), it starts a “breath holding” mode to limit its anaerobic respiration to the lowest level [70]. Flooding is serious abiotic stress affecting plant growth and can be classified into waterlogging and submergence. During the rainy season, the lotus is vulnerable to submergence. It is necessary to cultivate lotus varieties that are resistant to flooding to promote economic value. 

*WRKY* TFs play key roles in modulating plant biotic and abiotic stress response and secondary metabolic regulations. A total of 65 *WRKY* genes were identified in lotus, and they were regulated by salicylic acid (SA) and jasmonic acid (JA), of which *NnWRKY40a* and *NnWRKY40b* were significantly induced by JA and promoted benzylisoquinoline alkaloid (BIA) biosynthesis [71]. Lotus predominantly accumulates BIA, and the leaf and embryo have different alkaloid components that may be caused by two cluster *CYP80* genes synthetic bis-BIAs, and aporphine-type BIAs, respectively. Five TFs (3 *MYBs*, one ethylene-response factor, and one *bHLH*) were identified as the regulator involved in the BIA biosynthetic pathway in lotus [72]. 

## 5. Conclusions and Perspective

The new varieties of lotus with high yield, wide adaptability, and stress resistance play a vital role in improving the economic value of this important horticulture crop. The variations identification, functional gene cloning, and metabolites alterations among diverse germplasm resources were investigated in the past decades, driven by the progressively improved genome information which could facilitate breeding practices in lotus (Figure 1). However, a high-quality reference genome is the limiting factor that will affect the molecular breeding process. Improvement of the lotus reference genome will be a requisite in the future, directly affecting the accuracy of molecular markers and the efficiency of cloning functional genes. Gapless reference genomes and pan-genomes have become the new reference, based on which plentiful information of genomes such as open chromatin and more variant information can be explored. With the explosive growth of large-omics data, deep learning can be used to mine biological information and decipher gene regulation networks. Moreover, a sound genetic transformation system has not yet been well established in lotus, which still restricts the validation of gene function and genome-based gene editing, further hindering breeding strategies. Few studies on epigenomics, such as histone marks, accessible chromatin regions, and genomic interactions, have been conducted and are needed in future. Based on these investigations, collection of wild lotus germplasm and classification of both wild and cultivated germplasm, analysis of domestication, and identification of molecular markers and genes closely linked to important agronomic traits, should be systematically conducted in the coming years. Combining these and developing multiple breeding targets will speed up the breeding efficiency in lotus.

## Figures and Tables

**Figure 1 ijms-23-07270-f001:**
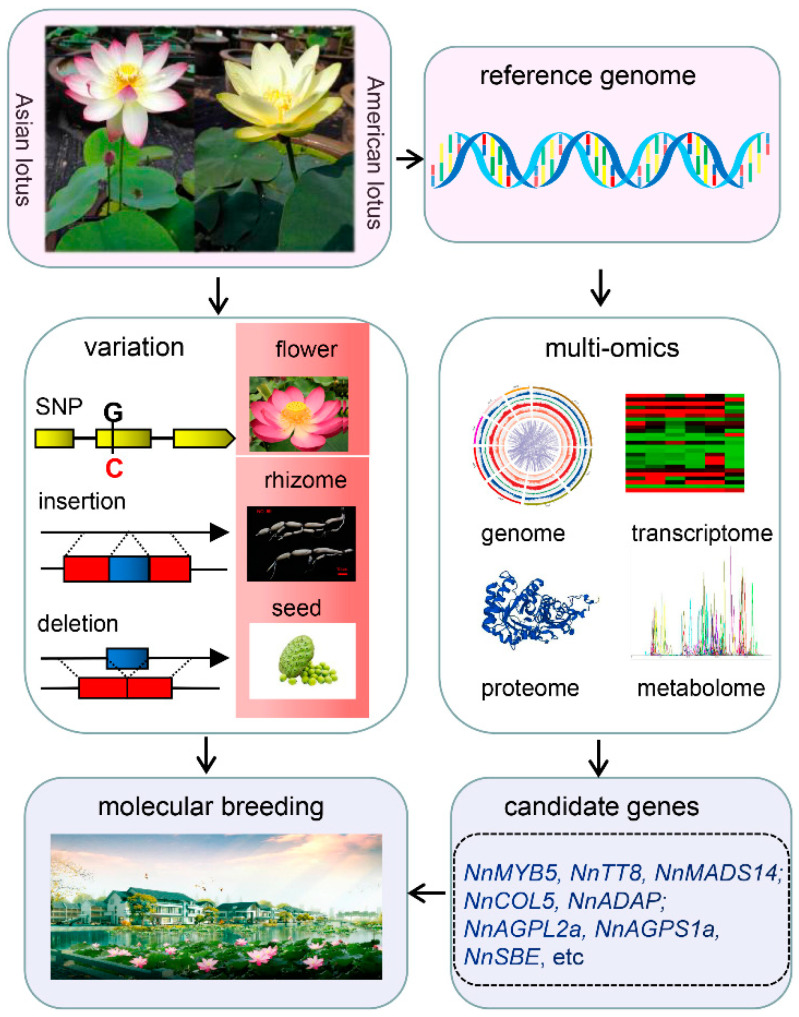
Flowchart of the molecular breeding process of lotus.

**Table 1 ijms-23-07270-t001:** Comparison of assembled lotus genomes.

Items	Year	Sequencing Technology	Final Assembly (Mb)	Contig N50	Number of Genes	Repeat Sequences	Ref.
China Antique v1.0	2013	Illumina, 454	804	38.8 Kb	26685	57%	[20]
Taizi	2013	Illumina Hiseq2000	792	39.3 Kb	40348	49.48% (TEs)	[21]
China Antique v2.0	2020	Pacbio Sequel, Illumina	821.2	484.3 Kb	32124	58.50%	[23]
Taikonglian NO.3	2022	Nanopore	807	5.1 Mb	28274	63.11%	[24]
American lotus	2022	Pacbio RSII, Hi-C	843	1.34 Mb	31382	81.00%	[25]

## Data Availability

Not applicable.

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
