# Peer review of "Studies on Lotus Genomics and the Contribution to Its Breeding"

_ijms, 2022, doi:10.3390/ijms23137270_

Round 1

Reviewer 1 Report

The review article entitled “Studies on lotus genomics and the contribution to its breeding” by Qi et al., present a comprehensive synthesis of the wealth of Lotus genome information. In addition, the authors describe a set of genes that can be relevant as traits for lotus breeding, according to the variability found between different lotus cultivars already sequenced. Finally, they pinpoint the future directions for the research of lotus aiming to potentially improve lotus breeding. Overall, this review is timely and relevant for the community of researchers interested in the improvement of plant breeding through the use of modern strategies that requires genomic data. Nevertheless, I think that the text should improve before publication. English professional editing is needed to polish the text. I have made some suggestions that can be found below:  

-What about including a comparison with the genome of another basal angiosperm? What is the economic impact of Lotus breeding?

-Information regarding the life cycle of lotus is missing.

-In Fig. 1, a picture of the complete plant (with scales) would be informative (not just the flower).

-Are there any lotus database for easy access? If so, please provide links in the text.

-More emphasis on the lack of epigenomic information is needed in the Conclusion. A plethora of epigenetic marks can be profiled in any plant species without the need for transformation, such as histone marks, accessible chromatin regions (ATAC-seq), genomic interactions (Hi-C).

Minor suggestions:

L12.: remove basic

L28-32: remove the sentences about the human genome project. Instead, you can mention something about the first plant and/or crop genomes (Arabidopsis, maize).

L35: a reference is missing.

L61: …many plants, such as? A few examples are expected.

L71-72: “…family, only…” (missing word in the middle; or change the punctuation of the sentence)

-It would be ok to add information about the complete geographical distribution of both Nelumbo species (or at least the Asian one).

L80: at/till

L93: …species IS playing… (missing verb)

L95: It would be nice to find more details about these unique features. What is flower thermogenesis? I’m not sure that the general community that reads IJMS is familiarized with these terms.

L97: what do you mean by “reality”? are there any unreal applications? Please improve!

Table 1: a column with the year of the release of each genome version would be informative.

-The 81.00% of TE in (26) is something to discuss further. Is significantly more than previous versions and is almost similar to maize.

L146: revise “showing”

L156: define SVs

L158: an/a

L175: indicated THAT the

L211: transcript factors…TRANSCRIPTION FACTORS???

L224: …has not well… (MISSING VERB)

L227: …gibberellic acid pathways and vernalization pathwayS

L231: What do you mean with ABUNDANT?

L313: salic acid (revise)

Author Response

The authors are grateful to this reviewer's valuable comments and suggestions. We have revised the manuscript accordingly. A native English speaker, Dr. Rebecca Damaris, who is our previous colleague, has helped us in the proof-reading and editing of the manuscript. Responses to comments were listed below point-by-point.

-What about including a comparison with the genome of another basal angiosperm? What is the economic impact of Lotus breeding?

Response: Thanks for this good suggestion. We have added the comparison with the genome of water lily (Nymphaea tetragona), which is an acient basal angiosperm. As we have mentioned in the background introduction, lotus might be the most important aquatic vegetable in China. Besides, lotus flower is among the top 10 famous Chinese traditional flowers, and it could also be used as traditional Chinese medicine. All these make it important to obtain high quality varieties through breeding.

-Information regarding the life cycle of lotus is missing.

Response: Thanks for this good suggestion. We have added this information in the revised manuscript.

-In Fig. 1, a picture of the complete plant (with scales) would be informative (not just the flower).

Response: As suggested, we modified the picture with the complete lotus plant on the left of the picture.

-Are there any lotus database for easy access? If so, please provide links in the text.

Response: Yes, there is a lotus database, which is accessible at (http://nelumbo.biocloud.net/nelumbo/home). We have provided this link in the text.

-More emphasis on the lack of epigenomic information is needed in the Conclusion. A plethora of epigenetic marks can be profiled in any plant species without the need for transformation, such as histone marks, accessible chromatin regions (ATAC-seq), genomic interactions (Hi-C).

Response: Thanks for this suggestion. Unfortunately, there are very few epigenetic studies on this species. As suggested, we have added the relevant content.

 Minor suggestions:

L12.: remove basic

Response: We have revised it.

L28-32: remove the sentences about the human genome project. Instead, you can mention something about the first plant and/or crop genomes (Arabidopsis, maize).

Response: We have revised it.

L35: a reference is missing.

Response: We have added the reference.

L61: …many plants, such as? A few examples are expected.

Response: We have added “such as rice, maize, Brassica and soybean”.

L71-72: “…family, only…” (missing word in the middle; or change the punctuation of the sentence)

Response: We have revised it.

-It would be ok to add information about the complete geographical distribution of both Nelumbo species (or at least the Asian one).

Response: We have added it.

L80: at/till

Response: We have revised it as “till”.

L93: …species IS playing… (missing verb)

Response: We have added it.

L95: It would be nice to find more details about these unique features. What is flower thermogenesis? I’m not sure that the general community that reads IJMS is familiarized with these terms.

Response: We have added the sentence “that may relate to flower protogyny or provide a warm environment for pollination [17,25]” after flower thermogenesis.

L97: what do you mean by “reality”? are there any unreal applications? Please improve!

Response: We have deleted “in reality”.

Table 1: a column with the year of the release of each genome version would be informative.

Response: We have added the information of the released year.

-The 81.00% of TE in (26) is something to discuss further. Is significantly more than previous versions and is almost similar to maize.

Response: The reason for high repeat sequence of American lotus is not mentioned in the literature. We added the sentence “It is interesting to investigate the dramatic difference of repeat sequence between them, because most of protein-coding genes in them show a one-to-one synteny pattern” in the text.

L146: revise “showing”

Response: We have revised the sentence as “Study of potential adaptive evolution and domestication of lotus”.

L156: define SVs

Response: We have defined SVs as structure variations.

L158: an/a

Response: We have revised it.

L175: indicated THAT the

Response: We have added it.

L211: transcript factors…TRANSCRIPTION FACTORS???

Response: We have revised it.

L224: …has not well… (MISSING VERB)

Response: We have added the “been” before “well” word.

L227: …gibberellic acid pathways and vernalization pathwayS

Response: We have revised it.

L231: What do you mean with ABUNDANT?

Response: We have revised it as “Lotus possesses distinct tpyes of the flower morphology”.

L313: salic acid (revise)

Response: We have revised it as “salicylic acid”.

Reviewer 2 Report

Dear authors, your work presented to me for review touches on a very interesting subject. However, it has some bugs that I recommend removing. This includes the introductory chapter. It contained unnecessary information not related to the topic of the work. I recommend removing them. In the introduction to scientific papers, the most important thing is to justify the topics that have been taken up. Therefore, information about the human genome as well as rice, tomato or soybeans is redundant and unnecessary. It is better to replace them by emphasizing the importance of the lotus as a useful plant for humans. How important this plant is and why. I have also posted comments on the manuscript. All the best.

Author Response

We are grateful to this reviewer's positive comments on our manuscript. As suggested, we have deleted most of the description on the genomic related studies on human, rice, tomato and soybean.

As for the comments listed in the text of manuscript, we have revised according to your suggestion. specifically, we described a little bit more about the number of varieties, and morphology of different types of varieties.